# Everyday Lives of Middle-Aged Persons with Multimorbidity: A Mixed Methods Systematic Review

**DOI:** 10.3390/ijerph19010006

**Published:** 2021-12-21

**Authors:** Ana Isabel González-González, Robin Brünn, Julia Nothacker, Christine Schwarz, Edris Nury, Truc Sophia Dinh, Maria-Sophie Brueckle, Mirjam Dieckelmann, Beate Sigrid Müller, Marjan van den Akker

**Affiliations:** 1Institute of General Practice, Goethe University, 60590 Frankfurt am Main, Germany; bruenn@allgemeinmedizin.uni-frankfurt.de (R.B.); dinh@allgemeinmedizin.uni-frankfurt.de (T.S.D.); brueckle@allgemeinmedizin.uni-frankfurt.de (M.-S.B.); dieckelmann@allgemeinmedizin.uni-frankfurt.de (M.D.); b.mueller@allgemeinmedizin.uni-frankfurt.de (B.S.M.); m.vandenakker@allgemeinmedizin.uni-frankfurt.de (M.v.d.A.); 2Health Services Research on Chronic Patients Network (REDISSEC), 28035 Madrid, Spain; 3Institute for Evidence in Medicine (for Cochrane Germany Foundation), Medical Center, Faculty of Medicine, University of Freiburg, 79106 Freiburg, Germany; nothacker@cochrane.de (J.N.); nury@ifem.uni-freiburg.de (E.N.); 4Instituto de Salud Carlos III, 28029 Madrid, Spain; christine_schwarz@hotmail.com; 5Department of Family Medicine, School CAPHRI, Maastricht University, 6200 Maastricht, The Netherlands; 6Academic Center for General Practice, Department of Public Health and Primary Care, KU Leuven, 3000 Leuven, Belgium

**Keywords:** multimorbidity, middle-aged, everyday life, coping skills, coping resources, systematic review

## Abstract

The healthcare burden of patients with multimorbidity may negatively affect their family lives, leisure time and professional activities. This mixed methods systematic review synthesizes studies to assess how multimorbidity affects the everyday lives of middle-aged persons, and identifies skills and resources that may help them overcome that burden. Two independent reviewers screened title/abstracts/full texts in seven databases, extracted data and used the Mixed Methods Appraisal Tool (MMAT) to assess risk of bias (RoB). We synthesized findings from 44 studies (49,519 patients) narratively and, where possible, quantitatively. Over half the studies provided insufficient information to assess representativeness or response bias. Two studies assessed global functioning, 15 examined physical functioning, 18 psychosocial functioning and 28 work functioning. Nineteen studies explored skills and resources that help people cope with multimorbidity. Middle-aged persons with multimorbidity have greater impairment in global, physical and psychosocial functioning, as well as lower employment rates and work productivity, than those without. Certain skills and resources help them cope with their everyday lives. To provide holistic and dynamic health care plans that meet the needs of middle-aged persons, health professionals need greater understanding of the experience of coping with multimorbidity and the associated healthcare burden.

## 1. Introduction

Multimorbidity, defined as the co-existence of two or more chronic diseases in the same individual, is an increasingly common global phenomenon [1]. Worldwide, more than 50% of middle-aged adults are multimorbid [2,3,4], and they may even be more numerous than elderly multimorbid adults overall [5]. Midlife is the time that chronic illnesses begin to surface, often taking adults by surprise, and negatively affecting their family lives, leisure time and professional activities [6]. 

Persons with multimorbidity often require multiple medications and specialized health care providers. Coordination of care is often lacking, and patients are at risk of poorer quality of life [7], higher health care costs [8] and a higher risk of mortality [9]. Furthermore, economic, social and psychological issues, as well as an individual’s decreasing functional capacity, reduced work productivity and working performance, may complicate the health management of middle-aged persons with multimorbidity [10].

Care for people with multimorbidity requires self-management and is often complex. Self-management involves taking actions and adopting behaviors to protect and promote health and manage the physical, psychological and social impact of having multiple chronic diseases [11]. Behaviors associated with the self-management of chronic conditions may include adhering to a medication regimen, monitoring illness symptoms, managing a diet, controlling weight, seeking regular health care and being physically active [11]. Self-management needs and capabilities affect everyday life and vary depending on the conditions.

Persons with multiple chronic conditions not only have to deal with the burden of the illness itself but also the burden of looking after themselves, which includes everything they have to do to manage their conditions. This burden may negatively affect the self-management of multimorbidity and influence health outcomes [12,13].

There is thus an urgent need for healthcare professionals and policy-makers to understand how multimorbidity affects this population. The aim of this study is to systematically review the literature to assess how multimorbidity affects the everyday lives (i.e., by focusing on family, leisure and work domains) of middle-aged persons and to identify skills and resources that can help them develop coping strategies to overcome the challenges of living with multimorbidity.

## 2. Materials and Methods

We have described the methodology in more detail in a study protocol (in press) and registered the systematic review in PROSPERO (registration no. CRD42021226699).

The current manuscript is written according to the Preferred Reporting System Items for Systematic Review and Meta-Analysis (PRISMA) checklist [14] (see Appendix A).

### 2.1. Design

We conducted a mixed methods systematic review and then used the convergent integrated approach to transform the data. This permitted us to simultaneously combine quantitative and qualitative data and simultaneously synthesize quantitative and qualitative studies [15].

### 2.2. Eligibility Criteria

#### 2.2.1. Types of Study

We included primary research studies that used quantitative (e.g., questionnaires), qualitative (e.g., focus groups) and mixed methods methodologies. We excluded case reports and articles that did not include a detailed description of methods, results or both, such as conference abstracts, narrative reviews, and editorials. 

#### 2.2.2. Participants

We included middle-aged persons (mean or median age 30–60 years) with multimorbidity (i.e., two or more simultaneous chronic conditions [1]). Studies focusing on persons with one chronic disease were included when authors had reported on at least one additional chronic condition in at least 80% of the study population. Studies that stratified results for middle-aged persons with multimorbidity were also included. 

We excluded studies that only addressed the experiences of caregivers, family members and health care professionals. 

#### 2.2.3. Outcomes

Everyday life domains of persons living with multimorbidity took into account their family lives, leisure time and career or work. Coping skills and resources in persons with multiple chronic conditions were also included. 

### 2.3. Information Sources and Search

We searched (from inception until 10 December 2020) the following electronic databases: MEDLINE; CINAHL; PsycINFO; Social Sciences Citation Index; Social Sciences Citation Index Expanded; PSYNDEX; and The Cochrane Library. We developed the final search strategy with an expert medical science librarian following PRESS Peer Review of Electronic Search Strategies recommendations [16]. The MEDLINE database search strategy used is provided in Appendix A. This strategy was adapted for use in the other databases. We did not apply any restrictions to publication date and included studies that were published in Dutch, English, French, German, Russian and Spanish.

We also examined the references of included studies, related systematic reviews and meta-analyses, and carried out forward citation tracking using the Web of Science Collection. 

### 2.4. Study Selection

All identified references were first imported to Endnote^©^ (Clarivate, Chandler, United States) and then uploaded to COVIDENCE^©^ (Covidence, Melbourne, Australia) for title, abstract and full-text screening. Duplicates were removed. Two reviewers (A.I.G.-G., R.B.) independently screened titles and abstracts to determine which should be assessed further. A stepwise calibration exercise was performed on a sample of 30 studies [17], with the aim of achieving at least 80% agreement between reviewers. The full texts of potentially eligible papers were then gathered and independently evaluated for eligibility by two reviewers (A.I.G.-G., R.B.). Any discrepancy was solved involving a third reviewer (M.v.d.A.).

### 2.5. Data Collection Process

One review author (A.I.G.-G.) extracted key study and participant characteristics and reported data on outcomes from all studies that fulfilled the inclusion criteria. A second review author (R.B.) crosschecked the extracted data. Any disagreement was solved with the help of a third author (M.v.d.A.). 

### 2.6. Data Items

We stratified the data according to study type (i.e., observational qualitative, quantitative and mixed methods, interventional) using standard extraction templates in Excel datasheets. Data were extracted and classified according to: (i) Study reference (i.e., first author, year of publication, title); (ii) Study aim; (iii) Study characteristics (i.e., study design, country of study origin, setting, sample size); (iv) Population characteristics (e.g., age, sex, definition of multimorbidity); (v) Data collection method (e.g., focus group or questionnaire); (vi) Description of outcomes (e.g., general functioning, physical, psychosocial and work functioning, coping skills and resources); and (vii) Results of described outcomes (e.g., proportion of participants whose work productivity had declined).

### 2.7. Risk of Bias 

One review author (A.I.G.-G.) used the Mixed Methods Appraisal Tool (MMAT) [18] to assess risk of bias (RoB). A second reviewer (R.B.) crosschecked. RoB assessment involved consultation with a third author (M.v.d.A.), when necessary. Studies were categorized according to study design, and the type of assessment depended on the employed methodology. 

### 2.8. Summary Measures

We provided summary statistics for each outcome, where possible. We reported study-level data narratively and visually (using graphs) and displayed the results in tabular form. 

### 2.9. Synthesis of Results

We conducted a mixed methods systematic review using a convergent integrated approach. We synthesized qualitative data by means of thematic synthesis, synthesized quantitative data and performed a meta-analysis where applicable, and, in a final step, integrated both qualtitative and quantitative synthesis according to the methodologies described by Sandelowski et al. [19], Pearson et al. [20] and the Joanna Briggs Institute [15]. 

For the qualitative analysis, two reviewers (A.I.G.-G., R.B.) independently analyzed the data and assigned thematic codes according to the above-mentioned classification of outcomes (i.e., global functioning, physical, psychological, social and work functioning, strategies used to cope). Both reviewers discussed coding and identified overarching thematic issues and categories using MAXQDA 18 software [21,22]. Disagreements were solved by discussion involving a third author (M.v.d.A.). 

For the quantitative analysis, data were analyzed descriptively. Meta-analysis of data was considered when the studies had comparable and sufficiently homogeneous outcomes. We first assessed heterogeneity qualitatively (in terms of study design, population and outcome), and then by means of X^2^ and additional tests. If meta-analysis was not possible, we carried out a descriptive analysis.

For the mixed methods synthesis, two reviewers (A.I.G.-G., R.B.) decided which format was the most promising and involved a third reviewer (M.v.d.A.) if consensus could not be reached. The decision depended mainly on the number of qualitative and quantitative studies that were eligible for inclusion. Wherever possible, qualitative data was converted into a numerical format for quantitative synthesis by using the Chang et al. approach [23] to transform verbal counts into numbers.

## 3. Results

After screening 7993 unique references, 44 studies were included in the systematic review (Appendix A). Appendix A presents excluded studies with reasons for exclusion.

### 3.1. Key Characteristics of the Included Studies and Participants

Table 1 and Table 2 show key characteristics of the included 44 studies and 49,371 patients (range 9–29,171). All studies used observational designs, 30 were quantitative, 13 were qualitative and 1 used mixed methods.

Of the 30 observational quantitative studies, 15 were conducted in North America and 13 in Europe. The studies were performed between 1994 and 2019, mostly using information from existing general population databases. Of the 13 qualitative studies, which were conducted in an outpatient setting from 2005 to 2019, 5 were carried out in North America and 4 in Europe. The number of patients included in the quantitative studies ranged from 32 to 29,171, and an overall 27,169 (58%) of them were female. The number of patients included in the qualitative studies ranged from 9 to 179 and an overall 257 of them were female (56%). Twelve studies included persons with multiple chronic conditions without any indication of an index disease, while the remaining thirty-two studies included persons with an index disease associated with at least one other chronic condition (i.e., cancer, cardiovascular disease, chronic kidney disease, diabetes, HIV, hypertension, mental health conditions, musculoskeletal disorders, neurological disorders and respiratory diseases). Mental health conditions accounted for 43% of the conditions.

### 3.2. Risk of Bias

The results of the RoB assessment are shown in Appendix A. 

All included studies had clear research questions and collected data that addressed the research questions. The 13 qualitative studies [27,41,45,46,47,49,51,55,57,60,61,63,64] had a low RoB. The quality of the 30 quantitative studies [24,25,26,28,29,30,31,33,34,35,36,37,38,39,40,42,43,44,48,50,52,53,54,56,58,59,62,65,66,67] varied, with 26 studies [24,25,28,29,30,31,33,34,35,36,37,38,42,44,48,50,52,53,54,56,58,59,62,65,67] providing insufficient information to assess response rate, 14 studies [25,36,37,38,39,40,43,44,52,53,54,58,62,65] providing insufficient information to assess representativeness and 9 studies [24,28,31,34,35,42,50,59,67] not providing a sample that was representative of the target population. The only mixed methods study [32] had a low RoB. 

All studies were included in the final synthesis, with greater emphasis placed on higher quality studies.

### 3.3. Everyday Life Domains of Persons Living with Multimorbidity

#### 3.3.1. Global Functioning

The two studies that addressed global functioning in middle-aged persons [30,65] used different scales to measure the extent to which multimorbidity had affected their day-to-day lives (Table 3). D’Amico et al. [30] discovered that persons with chronic migraines or medication overuse, headache and multimorbidity suffered more disability in domains such as cognition (i.e., understanding and communicating), mobility (i.e., moving and getting around), self-care (i.e., hygiene, dressing, eating and staying alone), interacting with others, life activities (i.e., domestic responsabilities, leisure, work and school) and participation (i.e., joining in community activities), than persons without multimorbidity. Wittchen et al. [65] considered handicaps in the areas of employment, family, social and romantic relationships, as well as the patient’s ability to perform routine Activities of Daily Living (ADL), and found that at least 50% of respondents indicated some current impairment due to social phobia.

#### 3.3.2. Physical Functioning

Fifteen studies [27,34,36,38,39,41,42,43,44,45,50,51,58,65,67] explored how physical functioning is affected in middle-aged persons with multimorbidity. 

Four studies described the impact the participants’ multiple illnesses and associated physical limitations had on their everyday lives [41,45,51,69]. In Noël et al. [45], the most frequently mentioned effects of multimorbidity on physical functioning were decreased mobility, physical limitations, pain and fatigue. Such physical limitations were also mentioned by some of the participants with multimorbidity in Sand et al. [51]. In Cheng et al. [27] the most common and serious issues reported by Chinese adults were physical limitations associated with their multiple chronic conditions. These were also described as limited mobility and pain. In Morgan et al. [41], it was common for women to describe the negative effect of their ill health on their ability to perform everyday tasks.

Romera et al. [50] assessed functional impairment with reference to the presence of painful physical symptoms in persons with generalized anxiety disorder, major depressive disorder or both. Of persons with both mental disorders, 78% had painful physical symptoms. Painful physical symptoms exacerbated functional impairment in all groups when compared to persons without, as reflected in a significantly higher total score on the Sheehan Disability Scale (SDS) [70]. The link between functional impairment and the presence of pain was relevant, as it was over 1.5-times worse when pain was present.

Egede et al. [34] and Tian et al. [58] assessed physical functioning in middle-aged persons with depression and associated comorbidities. Both studies showed that depression and comorbidities such as hypertension, diabetes mellitus and chronic obstructive pulmonary disease [34], or comorbid pain combined with other chronic diseases [58], resulted in greater limitations in ADL, such as walking, bathing, eating, dressing and getting in and out of bed. Deschênes et al. [67] showed that in individuals with diabetes, major depressive disorder or generalized anxiety disorder (or both) were associated with greater disability (e.g., getting out of bed) than in the control group.

Johs et al. [38] found that 18% of middle-aged persons with the Human Immunodeficiency Virus (HIV) and comorbidities reported at least one impairment in typical Instrumental Activities of Daily Living (IADL). Most of them reported difficulties with housekeeping (48%) and transportation (36%). Only 5% reported difficulties with medication management.

According to Gulley et al. [36] more than half of working-age people with functional disabilities reported having more than one chronic condition. Among those with ADL or IADL limitations, 35% reported having four or more chronic conditions.

We performed a meta-analysis based on seven [34,36,38,39,43,58,66] of the 15 studies mentioned above. The results showed that 29% (95% CI, 18–42%, I^2^ = 100%, *p* < 0.01) of the middle-aged persons with multimorbidity presented some kind of physical (functional) impairment that affected their ADL (Figure 1).

Two studies [42,44] measured functional disability using condition-specific measures. Motl et al. [42] found an association between physical activity (objectively measured by wearing an accelerometer and self-reported) and the number of self-reported cardiovascular comorbidities in persons with multiple sclerosis. Nikiphorou et al. [44] used the Bath Ankylosing Spondylitis Functional Index (BASFI) [71] to study the relationship between comorbidities and functional ability for spondylarthritis. In models adjusted for sociodemographic and clinical variables, a higher number of comorbidities, as measured using the Rheumatic Disease Comorbidity Index (RDCI) [72], was associated with higher BASFI. More specifically, an increase of one point in the RDI score was associated with an increase of approximately 0.37 points on the BASFI, indicating that higher comorbidity burden was associated with lower functional ability.

Table 4 shows the mean scores of both condition-specific physical disability scales for the two studies mentioned above [42,44].

#### 3.3.3. Psychosocial Functioning

Eighteen studies [25,26,27,31,33,41,43,45,47,48,51,52,54,55,59,63,65,67] explored psychosocial functioning in middle-aged persons with multimorbidity.

Diaz et al. [31] used the Supportive Care Needs Assessment Tool for Indigenous People (SCNAT-IP) [74] to investigate the association between comorbidity and supportive care needs among indigenous people in Australia that had been newly diagnosed with cancer. The most commonly reported moderate–high unmet needs were in the psychological (i.e., anxiety, feeling down or sad) and the practical and cultural (i.e., money worries) domains.

According to Noël et al. [45], the psychological reactions of middle-aged persons with multimorbidity comprised depression, anxiety, anger, irritability, inadequacy, resentment and feelings of loneliness and humiliation. Many participants expressed that their illnesses affected their ability to participate in or enjoy social and leisure activities. Others reported that their illnesses had negatively affected their family relationships or led to the break-up of their marriages. These findings were also confirmed by Cheng et al. [27], Morgan et al. [41] and Sand et al. [51].

Ørtenblad et al. [47] and White et al. [63] explored the burden of treatment among people with multimorbidity by investigating conflicts between leading their everyday lives and the burden of healthcare. They showed that multimorbid patients and their families often had to limit their social relationships in order to bear the treatment burden and adequately respond to family demands. 

Bell et al [25] measured the social adjustment of persons with dysthymia and comorbidities using the Social Adjustment Scale [75]. Persons with comorbid dysthymia had significantly lower scores in all areas of social functioning compared to persons without dysthymia. Based on an assessment of family functioning using the McMaster Family Assessment Device [76], a greater proportion of families with an adult with comorbid dysthymia were rated as dysfunctional, compared to families in which no adult had dysthymia (42 versus 8%, respectively).

Two studies [26,33] examined the role of alcohol use disorder in social functioning. In Buckner et al. [26], social anxiety disorder and alcohol use disorder were associated with significantly less social support from friends and greater stress in relationships with friends and relatives (but not partners) than social anxiety alone (Table 5, Table 6 and Table 7). In Dutton et al. [33], a combination of alcohol dependence and post-traumatic stress disorder led to greater apprehension and significantly less family support than in persons with only one or none of the conditions.

Deschenes et al. [67] assessed disability by asking patients with diabetes and a major depressive disorder, generalized anxiety disorder or both the question: “During the past two weeks, were there days on which you cut back on things because of illness?”. Individuals with diabetes, 12-month diagnoses of major depressive disorder without generalized anxiety disorder, generalized anxiety disorder without major depressive disorder or both major depressive disorder and generalized anxiety disorder showed greater social disability.

Saris et al. [52] found that depression and anxiety disorder are associated with greater impairment in behavioral (network size, social activities, social support) and affective (loneliness, affiliation, perceived social disability [77]) indicators of social functioning than healthy participants, or those with only one of the two diseases (Table 8 and Table 9). 

Neri et al. [43] reported severe limitations in social functioning according to the Short-Form Six-Dimension questionnaire [78] in 55% of persons with chronic kidney failure and multimorbidity. Rao et al. [48] showed that the likelihood of having difficulty interacting with others or maintaining a social life was higher in persons suffering from post-traumatic stress disorder and migraines compared to healthy controls or those with only migraines (Table 10).

In Schonauer et al. [54], the social networks of two deaf groups with psychiatric comorbidities (49 deaf schizophrenics and 38 deaf non-psychotics) were found to have greater shortcomings in the quantity and quality of social integration than those of non-deaf schizophrenic adults, with significant differences most obvious in the comparison between deaf schizophrenic and non-deaf schizophrenic adults. 

Slomka et al. [55] studied experiences of living with multimorbidity including HIV. The perception of a persistent stigma associated with HIV made participants reluctant to share information and/or to disclose the existence of HIV. In spite of long-term survival and the development of other potentially serious medical conditions, the stigma associated with HIV remained problematic for participants.

Todd et al. [59] used a retrospective case-control study. The group with mental health problems (e.g., substance misuse) plus a comorbidity showed higher levels of social exclusion on all reported measures (e.g., employment, homelessness, education) than those without a comorbidity. 

Wittchen et al. [65] showed how social phobia, whether comorbid, sub-threshold or pure, resulted in considerable subjective suffering and had a negative impact on social relationships.

#### 3.3.4. Work Functioning

Twenty-eight studies [24,27,28,29,30,34,35,37,39,40,41,43,44,45,47,48,49,50,51,53,54,56,58,59,62,63,65,66] assessed work functioning in middle-aged persons with multimorbidity.

Several participants in a study by Noël et al. [45] reported that their multiple chronic conditions had affected the quality of their job performance, or required them to change jobs or retire early.

Associations between mental disorders and work functioning were assessed in seven studies. Tian et al. [58] showed that persons with depression and comorbid pain were more likely to be unemployed or to have retired. Wittchen et al. [66] found that persons with depression and generalized anxiety disorder, and even more so in combination, were more impaired (defined as days limited in the past month) than those with neither disorder. Dagher et al. [29] showed that adults with a comorbidity of depression and substance abuse, dependence or both (SUD) were more likely to have undergone periods of unemployment, or to have been unemployed for 3 or more months, and and to have lower household income in midlife than those with neither disorder. Forty-five percent of this population had experienced a period of unemployment of 3 months or more in the past 10 years. 

Egede et al. [34] studied lost productivity (defined as days absent from work or spent in bed due to disease) in middle-aged persons with depression and comorbidities. Compared to those without, adults with chronic conditions and depression were more likely to spend at least 1 day per year in bed due to illness. Romera et al. [50] also studied lost productivity (defined as lost days and unproductive days) in persons with generalized anxiety disorders, major depressive disorder or both. The presence of painful physical symptoms was significantly associated with less productivity and a nearly two-fold increase in the number of underproductive days. Souêtre et al. [56] discovered that in employed persons with generalized anxiety disorders (79% of all participants), the prevalence of absenteeism from work was higher in those with comorbidities (34% versus 27%). However, the length of absenteeism did not differ between the two groups. Wittchen et al. [65] used the Work Productivity and Impairment Questionnaire (WPAI) to assess work productivity and found that 22% of respondents were unemployed and attributed the unemployment to their social phobia. Persons with comorbid social phobia had a significantly higher WPAI score than controls (12.4% versus 1.5% work productivity reduction).

Five studies assessed the association between work functioning and physical health conditions. Gehrke et al. [35] compared cancer survivors without to those with comorbidity. Those with were more likely to file discrimination claims due to the terms of their employment (OR 1.37, 95% CI 1.04–1.80) such as conditions, promotion, wages, and benefits. Hakola et al. [37] found that asthma and one other chronic condition increased the risk of long-term all-cause work disability with a hazard ratio (HR) of 2.2 (95% CI 1.78–2.83). Asthma and two or more further chronic conditions increased the HR to 4.5 (95% CI 1.62–6.78). Asthma with depression increased the risk to HR 3.6, with the risk of permanent work-disability especially high (HR 6.8). Nikiphorou et al. [44] used the the Work Productivity and Activity Impairment—Specific Health Problem (WPAI-SHP) questionnaire to study the relationship between comorbidities and work status (whether in employment or not) and productivity [79]. In adjusted models, for every unit increase in the RDCI, the likelihood of being in employment was reduced by 17%. This effect was similar to that of an increase in functional disability reflected in a higher BASFI. Higher RDCI was also significantly associated with higher levels of absenteeism and presenteeism. Schofield et al. [53] assessed the impact of comorbid conditions on labour force participation in persons with arthritis and found that the probability of being out of work increased in line with the number of comorbidities, and that 59% of those with arthritis and one or more comorbidities did not work. Weijman et al. [62] found that adults with diabetes and associated comorbidities were more likely to report fatigue-related complaints at work than adults with diabetes and no associated comorbidities, and than healthy employees. Furthermore, adults with diabetes and comorbidities worked more daytime hours and did less overtime than the other groups. 

Associations between work functioning and a combination of physical and mental disorders were assessed in five studies. Conover et al. [28] examined the labour market outcomes of adults with a combination of HIV, mental illness and substance abuse problems. While most private income stemmed from employment, less than 15% of this population was employed full or part-time. Compared with those in the best physical health, employment was lower among females and those in poor and moderate physical health. Li et al. [39] found that adults with rheumatoid arthritis and depression had more days of short-term disability than adults without depression, and were therefore less productive. Linder et al. [40] described difficulties in resuming work in those that were off work for long periods, and persons with psychiatric or somatic comorbidities. Eighty percent of those off work over the long term were assessed multidisciplinarily as in need of rehabilitation, whereby the rate was higher in persons with psychiatric diagnoses, with or without concomitant somatic diagnoses, than in persons with only somatic diagnoses.

Neri et al. [43] used the Work Ability Index (WAI) to explore the association between occupational stress and impaired work ability [80] in persons with comorbid chronic kidney failure. Occupational stress was measured using the Effort-Reward Imbalance (ERI) questionnaire [81]. Neri et al. found an inverse correlation between the ERI and the WAI (*p* < 0.04), even after adjusting for age, hourly income and comorbidity burden. Rao et al. [48] found that compared with healthy controls, those with post-traumatic stress disorder and migraines were more likely to live below the poverty line and less likely to have worked for pay or profit in the past week, but that those with either one or the other impairment were not. Additionally, the number of days on which work quality was lower due to physical or mental health or substance abuse in the past month was greater than among controls.

We included 19 [24,27,28,30,34,40,41,44,47,48,49,51,53,54,56,58,59,63,66] studies addressing employment in a meta-analysis, which showed that a mean of 44% (95% CI, 30–59%, I^2^ = 99%, *p* < 0.01) of middle-aged persons with multimorbidity were formally employed (Figure 2).

We performed a meta-analysis of two of the studies [65,66] and measured overall time at work in the past month. Twenty-four percent (95% CI, 16–34%, I^2^ = 0%, *p* = 0.87) of middle-aged persons with multimorbidity spent at least 50% less time working compared to persons without multimorbidity in the past month (Figure 3).

### 3.4. Coping Skills and Resources in Persons Living with Multimorbidity

Nineteen studies [24,25,26,28,32,41,46,47,49,51,54,55,57,60,61,63,64,69,82] assessed skills and resources that helped in the development of coping strategies to overcome the challenges of living with multimorbidity.

#### 3.4.1. Coping Skills

Bell et al. [25] used the Coping Response Inventory [83] to measure the coping ability of persons with comorbid dysthymia and compared it with that of persons who scored negatively due to any psychiatric disorder. Persons with comorbid dysthymia had significantly lower scores when having to cope with cognitive, behavioral, logical and general problems than persons without dysthymia. In addition, they had significantly higher scores in emotional discharge and avoidance coping, as well as decreased capacity for affective regulation.

Dickson et al. [32] showed that although several participants appeared overwhelmed by the management of their multiple chronic conditions and felt hopelessness, many more appeared to be resilient in spite of these challenges. Coping mechanisms that were mentioned included maintaining sense of humour, being active and responsible for their own care and counting with social support.

Cheng et al. [27] explored how Chinese adults cope with multiple chronic conditions. They found that people with multimorbidity are aware that good management of their illnesses allows them to continue to support their families. They also emphasized that they took responsibility for self-managing their illness for the sake of their familes rather than themselves.

O’Brien et al. [46] investigated the relationship between the management of multimorbidity and ‘everyday life work’ in patients living in areas of high socioeconomic deprivation in Scotland. Overall, most, and especially those with mental–physical multimorbidities, struggled with the amount of work required to establish a sense of normality in their everyday lives. However, they reported that daily chores gave them a sense of normality in their daily routines and helped them have fewer negative feelings, especially when they had lost their jobs due to their multiple illnesses.

Ørtenblad et al. [47] explored the burden of treatment in people with multimorbidity by investigating conflicts between everyday activities and health care needs. People with multimorbidity found that their treatment burden made it difficult to pursue their everyday lives as they would have liked. Difficulties were identified in three domains: (i) family and social life; (ii) work life; and (iii) goal-setting during appointments with health professionals. Individual resources and priorities in everyday life play a dominant role in resolving such difficulties and navigating conflicts between everyday activities and health care needs.

Sand et al. [51] explored the experiences of people with multimorbidity. The ability of people with multimorbidity to retain a sense of living a normal everyday life depended on how well they could deal with their multimorbidity. In the conflict between finding time for treatment appointments and for the family, sometimes the one and sometimes the other took priority. However, both patients and family members often cut back on their social relationships in order to be able to bear the treatment burden and to respond to family demands. 

Subramanian et al. [57] explored successful coping strategies used to overcome the life-altering changes in persons with chronic kidney disease and multimorbidity. Participants more commonly used multiple coping strategies that were related to more engagement. Frequent coping themes were “Take care of myself and follow doctors’ orders,” “accept it,” and “rely on family and friends”. The types of coping strategies used were influenced by factors such as treatment modality, time since diagnosis, presence of other chronic comorbidities, and self-perceived limitations.

White et al. [63] assessed the experience of living with multimorbidity. Participants described a evolving process that went from ‘recognizing something is not right,’ then ‘working out what is wrong,’ to later ‘getting things under control’ and finally ‘getting on with life’. 

White et al. [64] also explored the activities of adults with multimorbidity. Occupation was understood as performing meaningful activities within the context of one’s life, health conditions and environment. 

#### 3.4.2. Coping Resources

##### Financial Resources

Arnold et al. [24] explored how households cope with the economic burden of multimorbidity by asking them to choose from a range of financial resources such as income and savings, social welfare, support and donations from employers or agencies, support from social networks like friends and family, borrowing and raising money by selling assets. Twenty-four persons (75%) relied on income and savings, eighteen (56%) on social network support and seven (22%) on social welfare and donations (Table 11). 

Conover et al. [28] sought to determine the sources of income, including transfer income (e.g., welfare) and financial support from others of adults with a combination of HIV, mental illness and substance abuse problems. They showed that the average income of this population was below the poverty line for single individuals, and that more than two-thirds of their income came from public sources. The likelihood of receiving disability or retirement income was lower among those with the worst mental health.

Morgan et al. [41] explored the perceptions and experiences of women living with multimorbidity in Ghana. Women depended on the care and treatment provided by the health care system, despite inconsistent coverage and a lack of choice, and although their experiences varied depending on their chronic condition. Women depended on their family and community to offset the financial burden of treatment costs, and the situation was exacerbated by having many conditions. 

Schofield et al. [53] assessed the economic status of adults with arthritis with and without comorbidities. The private weekly income of those with arthritis and three or more comorbidities was 72% lower than in people with arthritis alone. They also paid less tax and received more government transfer payments.

##### Resources Used to Lessen Treatment (Drug) Burden

Ridgeway et al. [49] reported ways adults with multimorbidity use to lessen their treatment burden. The main themes gathered from the interviews: (i) problem-focused strategies such as self-care as part of their daily activities in life, asking others for support, future planning and use of technology; (ii) emotion-focused coping strategies such as having a positive attitude, thinking of other priorities in life and a focus on their own spirituality; (iii) social support; and (v) positive aspects of health care such as good coordination of care and relationships with professionals. Other subthemes that were observed from the focus groups included maintaining autonomy and dealing with healthcare professionals in a proactive way.

Slomka et al. [55] identified the problems that arose when managing medications in in persons with HIV and multiple chronic conditions. Besides monitoring medication effects, participants were encouraged to actively work with physicians in managing their care. 

Townsend et al. [60] examined attitudes towards the long-term use of medication in persons sufferring from four or more chronic diseases. Medication management occupied a central position in their lives, due to the complexity of the regimens and the strategies they had to adopt to be able of fulfill them. Almost half the respondents considered that following a complex medication regimen was the only way they could gain equilibrium, relief to their symptoms and experience a normal life. However, they referred that sometimes they had to perform changes to their fixed regimens for one or more of their diseases in order to adhere to their treatments. These changes were influenced by their experiences of symptoms in order to perform specific tasks and fulfil their social roles and obligations as grandparents, parents, employees, etc.

## 4. Discussion

The global, physical and psychosocial functioning ability of middle-aged persons with multimorbidity is lower than in non-multimorbid persons. They also have lower employment rates and lower productivity at work. In everyday life, middle-aged persons with multimorbidity cope by using a range of coping skills and resources. In general, we did not observe any pattern in terms of country or setting. Our study provides the first systematic review of how multimorbidity affects the everyday lives of middle-aged persons with multimorbidity by focusing on family, leisure and work domains, and the skills and resources that help them overcome the burden of treatment.

### 4.1. Summary of Evidence

In two studies [30,65], the global functioning of middle-aged persons with multimorbidity was shown to be worse than in persons without multimorbidity. Global functioning includes such domains as cognition, mobility, self-care, social activities, activities of daily living and employment.

In seven studies [34,36,38,39,43,58,66], one third of the middle-aged persons with multimorbidity had difficulty performing routine ADL, such as walking, bathing, eating, dressing or getting in and out of bed. Two studies [42,44] that measured physical functioning using condition-specific measures found that the extent of disability was associated with the number of chronic conditions. 

One study [45] showed that middle-aged persons with multimorbidity react psychologically to their impairment with, for example, depression, anxiety, anger, irritability, resentment and loneliness. Furthermore, the presence of depression and anxiety are associated with greater impairment in behavioral indicators of social functioning such as size of social network, social activities and social support, and in affective indicators such as loneliness, affiliation and perceived social disability, than healthy persons or persons with only one of the considered chronic conditions [52]. 

Nineteen studies [24,27,28,30,34,40,41,44,47,48,49,51,53,54,56,58,59,63,66] showed that only half the middle-aged persons with multimorbidity were in paid employment. It is important to consider an individual’s comorbidities when assessing the impact of a chronic condition on labour force participation and economic circumstances [28,29,53]. Labour force participation and economic circumstances are likely to be more irregular than among those with a single chronic disease [29,48,53]. 

Data from two studies that included persons with mental health conditions and multimorbidity showed that 24% had worked at least 50% less compared to persons without multimorbidity in the previous month. Cabral et al. [84] investigated the impact of multimorbidity on work through a literature review of longitudinal studies. They concluded that multimorbidity has a negative impact on work, with a knock-on effect on quality of life, work productivity and absenteeism and presenteeism indices. This increases the likelihood of temporary and permanent sick leave, and lowers employability.

Mental health conditions accounted for almost half the identified index diseases included in this review. Furthermore, studies that included depression [29,34,36,39,50,52,58,66,67] found an association with greater disability, whereby the number of depressive symptoms was associated with the extent of the limitations in social and general activities, and linked to an increase in physical impairment. Depression therefore appears to be both a risk factor for and a consequence of multimorbidity that exacerbates disability. The association between multimorbidity and depression has been described as being bidirectional, with depression increasing the risk of multimorbidity and vice versa [85]. This association was also found in middle-aged Chinese patients [86] and may be the result of symptom overlap, such as sleep and eating disorders and increased fatigability [87].

Persons with multimorbidity experience difficulties organizing their everyday lives and their health treatment to take into account their individual priorities [47]. Living with a chronic illness involves recognizing when something is not right, working out what is wrong, getting things under control and getting on with life [63]. Liddy et al. [88] identified in a systematic review of qualitative literature that the important themes raised by people with multimorbidity included living with undesirable physical and emotional symptoms, and particularly with pain and depression. The use of cognitive strategies, including reframing, prioritizing, and changing beliefs, was reported to improve people’s ability to self-manage their multiple chronic conditions. In a systematic review, Rosbach et al. [89] found that people with multimorbidity appear to use strategies such as prioritizing certain treatments to diminish their workload and mobilize and coordinate resources to improve their ability to manage their treatment burden. They try to routinize their treatment and integrate it into their daily lives, presumably in order to maintain a balance between workload and capacity.

According to one study [24], the majority of middle-aged persons with multimorbidity relied largely on their own income and savings to cope with the economic burden of multimorbidity. However, more than half relied on support from their social networks. Besides, the average income of this population was lower than in persons with just one chronic disease and was sometimes below poverty levels [28,82]. Sum et al. [90] showed that out-of-pocket expenditure, often on medicines, can severely compromise an individual’s financial situation, and that non-adherence to prescribed drug regimens was found to be a coping strategy used to limit out-of-pocket expenditure on medicines.

### 4.2. Limitations

Several limitations of our review should be considered when interpreting the results.

First, the review included studies with varying characteristics and populations. Such methodological differences (e.g., data collection methods, different measures and scales), and clinical differences (e.g., type and severity of conditions) may partly explain variations in our findings. Most findings appeared to be disease specific (e.g., depression) and not valid for all combinations of chronic conditions.

Second, the relatively small sample sizes for many of the outcomes may have limited the generalizability of our results (e.g., low productivity). 

Third, the individual studies assessed a convenience sample of participants and did not always use a control group to compare outcomes. 

Fourth, all studies were cross-sectional, thus preventing the identification of causal and temporal patterns. Relationships between assessed outcomes and multimorbidity could therefore not be ascertained. Besides, most data were self-assessed, which may have biased estimates in both directions. 

Fifth, over half the studies provided insufficient information to assess representativeness or response bias, which did not allow us to assess a potential selection bias.

Finally, we did not include the experiences of caregivers, family members and health care professionals, which may contribute to the way multimorbidity affects the everyday lives of middle-aged persons, as well as their ability to cope with the challenges of living with multimorbidity. 

## 5. Conclusions

This review assesses how multimorbidity affects the everyday lives of multimorbid persons and the skills and resources they rely on to develop coping strategies. Healthcare professionals should be aware that the disease burden of middle-aged persons with multimorbidity may be particularly dependent on their physical, financial, and interpersonal (family and social network) circumstances. Furthermore, mental health has shown to be a relevant factor that should be taken into account when managing middle-aged persons with multimorbidity. It is therefore important that professionals not only understand how multimorbidity affect middle-aged persons’ lives but also to consider their experiences of coping with multimorbidity so they can provide holistic and dynamic health care plans that are tailored according to their actual needs. 

## Figures and Tables

**Figure 1 ijerph-19-00006-f001:**
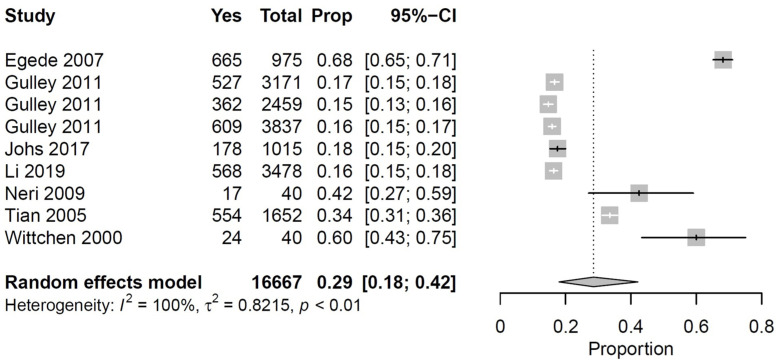
Forest plot shows the percentage of middle-aged persons with multimorbidity and physical impairment (of any kind). Gulley et al. shows results for three different subgroups of that population corresponding (from top to bottom) to persons with arthritis, diabetes and depression.

**Figure 2 ijerph-19-00006-f002:**
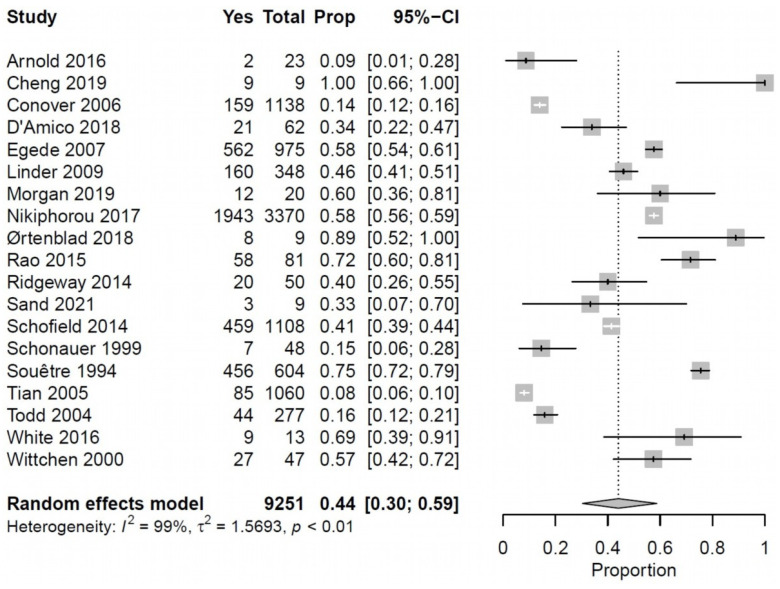
Forest plot shows the percentage of middle-aged persons with multimorbidity and paid employment. When retired populations were identified, they were excluded.

**Figure 3 ijerph-19-00006-f003:**
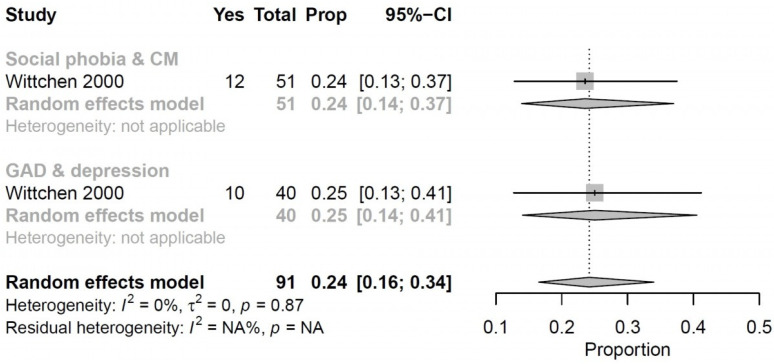
Forest plot shows the percentage of middle-aged persons with multimorbidity that in the past month had worked at least 50% less than those that were not multimorbid.

**Table 1 ijerph-19-00006-t001:** Key characteristics of the included studies.

Author, Year (Reference)	Country	Setting	Design	Data Collection Methods	Response (%)	Participants *n*	Female Sex, %	Age Mean, SD	Condition	Everyday Life Domains/Coping Strategies under Review
Arnold, 2016 [24]	Kyrgyzstan	Mixed-general practice and outpatient (specialized)	Observational, quantitative	Questionnaires	nr	32	38	54, nr	Diabetes and TB	Work functioning and coping resources (financial)
Bell, 2004 [25]	Canada	General practice	Observational, quantitative	Interviews	77	172	71	42, 14	Dysthymia and CM	Social functioning and coping
Buckner, 2008 [26]	USA	Community	Observational, quantitative	Interviews	83	195	40	33, 11	Social anxiety disorder and alcohol dependence	Social functioning and coping
Cheng, 2019 [27]	China	Outpatient (specialized)	Observational, qualitative	Semi-structured interviews	nr	14	43	53, 14	MM	Physical, psychosocial and work functioning, coping
Conover, 2006 [28]	USA	Community	Observational, quantitative	Interviews	nr	1138	58	38, nr	HIV, chronic mental illness and SUD	Work functioning and coping resources (financial)
Dagher, 2015 [29]	USA	Community	Observational, quantitative	Survey	nr	66	39	32, 0.5	SUD and depression	Work functioning
D’Amico, 2018 [30]	Italy	Outpatient (specialized)	Observational, quantitative	Interviews	nr	62	77	48, nr	MM (including MOH or migraines)	Physical, psychosocial and work functioning
Deschenes, 2015	Canada	Community	Observational, quantitative	Interviews	nr	145	nr	nr	Diabetes, depression and/or GAD	Physical and social functioning
Díaz, 2016 [31]	Australia	Outpatient (specialized)	Observational, quantitative	Questionnaires	nr	107	55	57, 11	Cancer and CM	Social functioning
Dickson, 2013 [32]	USA and Australia	Outpatient (specialized)	Observational, mixed methods	Questionnaires	nr	114	38	59, 15	MM (including heart failure)	Coping
Dutton, 2014 [33]	USA	Community	Observational, quantitative	Interviews	nr	56	57	41, 1	PTSD and alcohol dependence	Social functioning
Egede, 2007 [34]	USA	Community	Observational, quantitative	Questionnaires	nr	975	64	46, nr	Depression and CM	Physical and work functioning
Gehrke, 2017 [35]	USA	Community	Observational, quantitative	Survey	nr	333	63	49, 9	Cancer and CM	Work functioning
Gulley, 2011 [36]	USA	Community	Observational, quantitative	Survey	nr	29,171	57	45, 0.1	Arthritis, diabetes and depression	Physical functioning
Hakola, 2011 [37]	Finland	Community	Observational, quantitative	Survey	74	2332	85	45, 10	Asthma and CM	Work functioning
Johs, 2017 [38]	USA	Outpatient (specialized)	Observational, quantitative	Questionnaire	nr	1015	19	51, 8	HIV and CM	Physical functioning
Li, 2019 [39]	USA	Community	Observational, quantitative	Survey	100	3478	87	51, 10	Rheumatoid arthritis and depression	Physical and work functioning
Linder, 2009 [40]	Sweden	Community	Observational, quantitative	Questionnaire	nr	348	63	46, 8	Psychiatric and somatic diagnosis	Work functioning
Morgan, 2019 [41]	Ghana	Outpatient (specialized)	Observational, qualitative	Interviews	nr	20	100	55, 10	MM	Physical, social and work functioning, coping
Motl, 2011 [42]	USA	Outpatient (specialized)	Observational, quantitative	Questionnaires	nr	561	83	47, 10	Multiple sclerosis and CVD	Physical functioning
Neri, 2009 [43]	USA	Outpatient (specialized)	Observational, quantitative	Questionnaires	87	40	25	47, 8	CKD and CM	Physical, social and work functioning
Nikiphorou, 2017 [44]	Multi-national	Outpatient (specialized)	Observational, quantitative	Questionnaires	nr	3370	34	43, 14	Spondylarthritis and CM	Physical and work functioning
Noël, 2005 [45]	USA	General practice	Observational, qualitative	Focus groups	77	60	20	50, nr	MM	Physical, psychosocial and work functioning
O’Brien, 2014 [46]	UK	General practice	Observational, qualitative	Semi-structured interviews	nr	14	50	54, 5	MM	Coping
Ørtenblad, 2018 [47]	Denmark	Outpatient (specialized)	Observational, qualitative	Focus groups	nr	10	50	51, 8	MM	Social and work functioning, coping
Rao, 2015 [48]	USA	Community	Observational, quantitative	Survey	nr	68	81	40, 1	Migraines and PTSD	Social and work functioning
Ridgeway, 2014 [49]	USA	Outpatient (specialized)	Observational, qualitative	Interviews and focus groups	nr	50	58	54, 13	MM	Work functioning and coping
Romera, 2011 [50]	Spain	General practice	Observational, quantitative	Questionnaire	nr	559	77	52, 15	GAD and depression	Physical and work functioning
Sand, 2021 [51]	Denmark	General practice	Observational, qualitative	Semi-structured interviews	nr	9	66	54, 9	MM	Physical, psychosocial and work functioning, coping
Saris, 2017 [52]	Netherlands	Outpatient (specialized)	Observational, quantitative	Survey	nr	748	68	41, 12	Anxiety and depression	Social functioning
Schofield, 2014 [53]	Australia	Community	Observational, quantitative	Survey	nr	1108	nr	53, nr	Arthritis and CM	Work functioning and coping resources (financial)
Schonauer, 1999 [54]	Germany	Outpatient (specialized)	Observational, quantitative	Semi-structured interview (sign language)	nr	49	33	37, 11	Prelingual deafness and Schizophrenia	Social and work functioning, coping
Slomka, 2017 [55]	USA	Outpatient (specialized)	Observational, qualitative	Focus groups	nr	22	27	51, nr	HIV and CM	Social functioning and coping
Souêtre, 1994 [56]	France	General practice	Observational, quantitative	Questionnaires	nr	604	60	41, 14	GAD and CM	Work functioning
Subramanian, 2017 [57]	USA	Community	Observational, qualitative	Semi-structured interviews	99	179	55	57, 7	CKD and CM	Coping
Tian, 2005 [58]	USA	Community	Observational, quantitative	Survey	nr	1652	65	60, 0.2	Depression and CM	Physical and work functioning
Todd, 2004 [59]	UK	Outpatient (specialized)	Observational, quantitative	Survey	nr	277	26	37, 11	Mental health problems and SUD	Social and work functioning
Townsend, 2003 [60]	UK	Community	Observational, qualitative	Interviews	56	23	57	50, nr	MM	Coping
Warren-Jeanpiere, 2014 [61]	USA	Outpatient (specialized)	Observational, qualitative	Focus groups	46	23	100	57, 3	HIV and CM	Coping
Weijman, 2004 [62]	Netherlands	Community	Observational, quantitative	Questionnaires	nr	65	17	49, 7	Diabetes and CM	Work functioning
White, 2016# [63]	Australia	Outpatient (specialized)	Observational, qualitative	Interviews	100	16	69	47, 12	MM	Social, work functioning and coping
White, 2019# [64]	Australia	Outpatient (specialized)	Observational, qualitative	Interviews	100	16	69	47, 12	MM	Coping
Wittchen, 2000 [65]	Germany	Community	Observational, quantitative	Interviews	nr	51	61	38, 10	Social phobia and CM	Psychosocial and work functioning
Wittchen, 2000 [66]	Germany	Community	Observational, quantitative	Interviews	89	40	52	40, nr	GAD and depression	Physical and work functioning

CKD = Chronic Kidney Disease; CM = Comorbidities; CVD = Cardiovascular Disease; GAD = Generalized Anxiety Disorder; MOH = Medication Overuse Headache; MM = Multimorbidity; nr = not reported; PTSD = Posttraumatic Stress Disorder; SUD = Substance Abuse and/or Dependence; TB = Tuberculosis. # Same population.

**Table 2 ijerph-19-00006-t002:** Descriptive summary of included studies.

Variable	Total—*n* (%)
Study characteristics	
Geographical location *	
- North America	21 (48%)
- Europe	17 (35%)
- Australia	5 (10%)
- Asia	1 (2%)
- Africa	1 (2%)
Setting	
- Community	19 (43%)
- Outpatient (specialized)	17 (39%)
- General practice	6 (14%)
- Mixed	2 (5%)
Design	
- Quantitative	30 (68%)
- Qualitative	13 (30%)
- Mixed methods	1 (2%)
Data collection methods *	
- Questionnaire/survey	22 (50%)
- Interview	13 (30%)
- Semi-structured interview	5 (11%)
- Focus group	5 (11%)
Sample size—Total (range) †	49,371 (9–29,171)
- Observational	
○ Quantitative †	48,817 (32–29,171)
○ Qualitative	440 (9–179)
○ Mixed methods	114 (59)
Patients’ characteristics	
Type of condition *	
- Studies describing patients with multimorbidity	12 (27%)
- Studies describing patients with a chronic condition associated with multimorbidity	
○ Cancer	2 (5%)
○ Cardiovascular disease	1 (2%)
○ Chronic kidney disease	2 (5%)
○ Diabetes	5 (11%)
○ HIV	4 (9%)
○ Hypertension	1 (2%)
○ Mental health conditions	19 (43%)
○ Musculoskeletal disorders	4 (9%)
○ Neurological disorders	2 (5%)
○ Respiratory diseases	1 (2%)
Age (range)	32–60
- Early middle age (30–44)	13 (30)
- Late middle age (45–65)	30 (68)
Sex (% female)	28,477 (58%)

* Studies may be included in more than one category; † Studies including the same participants were counted only once [63,64].

**Table 3 ijerph-19-00006-t003:** Global functioning impairment.

First Author, Year	Type of Condition	Age, Mean (SD)	Participants, *n*	Scale Name	Score
D’Amico, 2018 [30]	MM	48 (12)	62	WHODAS—mean (SD) *	35 (13)
Wittchen, 2000 [65]	Social phobia and CM	38 (10)	51	Disability Self-Rating Scale—LDSRS—mean [68] **	67.9

*n* = number; MM = Multimorbidity; * WHODAS 2.0 assesses disabilities in the domains of cognition, mobility, self-care, interacting with others, life activities and participation. Summary score ranges from 0 to 100, where 0 = no disability and 100 = full disability. ** The LDSRS scale measures impairment due to emotional problems in the areas of education, employment, family, social and romantic relationships, and the patient’s ability to perform routine activities of daily life. It comprises 11 items, each scored from ‘0’ (no effect) to ‘3’ (severe limitation). A maximum score of 100% means no impairment; a score of 39% means substantial disability.

**Table 4 ijerph-19-00006-t004:** Condition-specific physical impairment.

First Author, Year	Type of Condition	Age, Mean (SD)	Participants, *n*	Scale Name	Score
Motl, 2011 [42]	Multiple sclerosis and CVD	47 (12)	561	Patient-Determined Disease Steps (PDDS) scale [73]—Mean (SD) *	3 (1)
Nikiphorou, 2017 [44]	Spondylarthritis and CM	43 (14)	3349	Bath Ankylosing Spondylitis Functional Index (BASFI) [71]—Mean (SD) **	3 (3)

CM = Comorbidities; CVD = Cardiovascular Disease; *n* = number; SD = Standard Deviation. ***** The PDDS scale is a self-report questionnaire that contains a single item for measuring self-reported disability using an 8-level ordinal scale that ranges from 0 = ‘normal’ to 8 = ‘bedridden’ or ‘unable to sit in a wheelchair for more than one hour’. ** Ranging 0–10, with a higher score indicating more disability.

**Table 5 ijerph-19-00006-t005:** Social anxiety-related impairment.

First Author, Year	Type of Condition	Age, Mean (SD)	Participants, *n*	Score Mean (SD) *
Buckner, 2008 [26]	Social anxiety disorder and alcohol dependence	33 (11)	195	4 (2)

*n* = number. * Patients rated the extent to which their social anxiety fear(s) ever interfered with their activities, and the degree to which avoidance of social anxiety-provoking situations interfered. Interference was assessed with a 4-point Likert scale (from significantly to not at all) in both cases. The sum of these two items produced the total social anxiety-related impairment score.

**Table 6 ijerph-19-00006-t006:** Relationship stress.

First Author, Year	Type of Condition	Age, Mean (SD)	Participants, *n*	Outcome Categories	Score Mean (SD) *
Buckner, 2008 [26]	Social anxiety disorder and alcohol dependence	33 (11)	195	Partner	15 (4)
				Relatives	15 (4)
				Friends	14 (4)

*n* = number. * Possible scores on each scale ranged from a low of 6 (reflecting minimal conflict and stress) to a high of 24 (reflecting high stress and conflict).

**Table 7 ijerph-19-00006-t007:** Social functioning I–Perceived social support.

First Author, Year	Type of Condition	Age, Mean (SD)	Participants, *n*	Scale Name	Score Mean (SD) *
Buckner, 2008 [26]	Social anxiety disorder and alcohol dependence	33 (11)	195	Partner	22 (3)
				Relatives	19 (4)
				Friends	19 (5)

*n* = number; SD = Standard Deviation. * Possible scores ranged from a minimum of 6 (reflecting minimal perceived support) to a maximum of 24 (reflecting high perceived support).

**Table 8 ijerph-19-00006-t008:** Social functioning II.

First Author, Year	Type of Condition	Age, Mean (SD)	Participants, *n*	Scale Name	Score Mean (SD)
Saris, 2017 [52]	Anxiety and depression	41 (12)	748	Network size *	2 (1)
				Social activities ^$^	12 (4)
				Social support ^#^	25 (13)

*n* = number; SD = Standard Deviation. * Six-point scale [1 (0 or 1 individuals in their network), 2 (2–5 individuals), 3 (6–10 individuals), 4 (11–15 individuals), 5 (16–20 individuals) and 6 (>20 individuals)] that indicates the number of adults with whom the patient has regular and important contact. ^$^ Frequency of occurrence of five social activities such as cultural events, trips to the countryside, visits to restaurants, social meetings and outdoor sporting activities, with a range that goes from almost never (1) to several times per week (6). A sum score (range from 5 to 30) is calculated by adding the frequency the five social activities. ^#^ The Close Persons Questionnaire (CPQ) measures the amount of social support that participants receive.

**Table 9 ijerph-19-00006-t009:** Socio-affective functioning II.

First Author, Year	Type of Condition	Age, Mean (SD)	Participants, *n*	Scale Name	Score Mean (SD)
Saris, 2017 [52]	Anxiety and depression	41 (12)	748	Loneliness scale [73] *	7 (4)
				Affiliation ^$^	4 (2)
				WHO-Disability Assessment Schedule, 2nd version (WHODAS 2.0) [77] ^#^	14 (5)

*n* = number; MM = Multimorbidity; SD = Standard Deviation. * The Jong Gierveld Loneliness Scale assesses feelings of loneliness based on 11 questions. ^$^ Affiliation is measured using the 6-item self-report “need for affiliation” scale. ^#^ WHODAS 2.0 assesses disabilities in the domains of cognition, mobility, self-care, interacting with others, life activities and participation. This study assessed only interactions with others and referred to social relations and difficulties that might be encountered with this life domain due to a health condition; in this context, “other people” included those known well (e.g., spouse or partner, family members or close friends) and those that were not (e.g., strangers).

**Table 10 ijerph-19-00006-t010:** Social affective functioning II.

First Author, Year	Type of Condition	Age, Mean (SD)	Participants, *n*	Outcome	*n* (%)
Neri, 2009 [43]	CKD and CM	47 (8)	40	Severely limited social functioning	22 (55)
Rao, 2015 [48]	Migraines and PTSD	33 (11)	68	Difficulty interacting with others or maintaining a social life due to health problems	27 (39)

CKD = Chronic Kidney Disease; *n* = number; PTSD = Posttraumatic Stress Disorder; SD = Standard Deviation.

**Table 11 ijerph-19-00006-t011:** Financial coping strategies.

First Author, Year	Type of Condition	Age, Mean (SD)	Participants, *n*	Scale Name	*n* (%)
Arnold, 2016 [24]	Diabetes and TB	54 (NA)	32	Number of coping strategies, mean (95% CI)	1.66 (1.32–1.99)
				income and savings	24 (75)
				social welfare and donations	7 (22)
				social networks support	18 (56)
				borrowing money	3 (9)
				selling household assets	1 (3)

CI = Confidence Interval; *n* = number; NA = Not Available; TB = Tuberculosis.

## Data Availability

Data is contained within the article or Appendix A.

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
