# Peer review of "Everyday Lives of Middle-Aged Persons with Multimorbidity: A Mixed Methods Systematic Review"

_ijerph, 2021, doi:10.3390/ijerph19010006_

Round 1

Reviewer 1 Report

The article on systematic review including qualitative and quantitative studies is very well written and met the requirements of the systematic reviews using PRISMA. 

  1. There is a new version of PRISMA, suggest to use that version. I think the difference is in the identification of references section. (starting...)
  2. Evidence of the quality of the references were not discussed.
  3. Most impressive was the meta analysis (I suggest to include the p-value in the description of the text taken from the plot)
  4. Conclusion can be improved as it does not give an impressive reading  of a good work shown in this article. 

Author Response

Revisions (R) made according to BMJ Open reviewer's report (Reviewer 1) by queries (Q):

Q1 - The article on systematic review including qualitative and quantitative studies is very well written and met the requirements of the systematic reviews using PRISMA. 

R1 – We appreciate the positive comments from Reviewer 1.

Q2 - There is a new version of PRISMA, suggest to use that version. I think the difference is in the identification of references section. (starting...)

R2 – We believe we used the last version of both the checklist (PRISMA 2020 checklist) and the flow diagram (PRISMA 2020 flow diagram form new systematic reviews which included searches of databases and other sources) which can be found at the following link: http://www.prisma-statement.org/PRISMAStatement/FlowDiagram

Q3 - Evidence of the quality of the references were not discussed.

R3 – We agree with reviewer 1 and included a paragraph in the section 4.2. Limitations (lines 692-694) as follows:

“Fifth, over half the studies provided insufficient information to assess representativeness or response bias, which did not allow us to asses a potential selection bias.”

Q4 - Most impressive was the meta analysis (I suggest to include the p-value in the description of the text taken from the plot)

R4 – We included the p-value in the description of the text taken from the plots as per reviewer 1 suggestions.

Q5 - Conclusion can be improved as it does not give an impressive reading of a good work shown in this article. 

R5 – We improved the conclusion by adding the following text in to the conclusion section (lines 704-709):

“Furthermore, mental health has shown to be a relevant factor that should be taken into account when managing middle-aged persons with multimorbidity. It is therefore important that professionals not only understand how multimorbidity affect middle-aged persons’ lives but also to consider their experiences of coping with multimorbidity so they can provide holistic and dynamic health care plans that are tailored according to their actual needs.”

Reviewer 2 Report

I would like to express my deepest gratitude for the opportunity to review the paper. I respect that the authors have taken so much time and effort to conduct this huge review study.

In this study, the authors sought to identify the impact of multimorbidity (MM) on patients' everyday lives and the skills and resources they coped with it. As a result of their research, the authors noted that the burden of disease in patients with MM is particularly dependent on their physical, financial, and interpersonal circumstances.

The authors cite as their 15th reference the following value of the mixed methods approach. (I recommend that the authors include the URL of this document)

//////////////////////
In this example, the knowledge gained from the qualitative and economic evidence can be used to enhance the knowledge gained from the quantitative evidence.

https://www.researchgate.net/publication/319713049_2017_Guidance_for_the_Conduct_of_JBI_Scoping_Reviews
//////////////////////

The emphasis here is on the synergistic effect of quantitative and qualitative research working in complementary ways. 

However, I am concerned that the authors do not adequately integrate the results of previous studies. The results of the present study were merely a list of previous studies.

The reason for this is that the aim of the study was to focus on MM, which is defined by the number of diseases, whereas the results by the authors focused on the contents of the patients' disease.

Are the authors interested in the number of diseases? Or is it the contents of the patient's disease?

Lastly, in order to brush up on this study, the framework of meta-epidemiological methodology research may be helpful because all the studies included in this study were observational studies.

Author Response

Revisions (R) made according to BMJ Open reviewer's report (Reviewer 2) by queries (Q):

Q6 - I would like to express my deepest gratitude for the opportunity to review the paper. I respect that the authors have taken so much time and effort to conduct this huge review study. In this study, the authors sought to identify the impact of multimorbidity (MM) on patients' everyday lives and the skills and resources they coped with it. As a result of their research, the authors noted that the burden of disease in patients with MM is particularly dependent on their physical, financial, and interpersonal circumstances.

R6 – We thank Reviewer 2 for the encouraging appreciation and the well-informed, pertinent comments.

Q7 - The authors cite as their 15th reference the following value of the mixed methods approach. (I recommend that the authors include the URL of this document). In this example, the knowledge gained from the qualitative and economic evidence can be used to enhance the knowledge gained from the quantitative evidence.

https://www.researchgate.net/publication/319713049_2017_Guidance_for_the_Conduct_of_JBI_Scoping_Reviews

The emphasis here is on the synergistic effect of quantitative and qualitative research working in complementary ways. However, I am concerned that the authors do not adequately integrate the results of previous studies. The results of the present study were merely a list of previous studies.

R7 – We included the URL of reference #15 as suggested by reviewer 2. As mentioned in the manuscript provided by reviewer 2, it is important to highlight the distinction between scoping reviews and “mixed methods” systematic review. We aimed to conduct a systematic review that allowed us “to assess how multimorbidity affects the everyday lives (i.e., by focusing on family, leisure and work domains) of middle-aged persons and to identify skills and resources that can help them develop coping strategies to overcome the challenges of living with multimorbidity”. Therefore, our aim was not to map or chart the evidence but to find actual answers in the literature. Unfortunately, the heterogeneity of the studies found, as mention in the discussion section, did not allow us to adequately integrate the results.

Q8 - The reason for this is that the aim of the study was to focus on MM, which is defined by the number of diseases, whereas the results by the authors focused on the contents of the patients' disease. Are the authors interested in the number of diseases? Or is it the contents of the patient's disease?

R8 – Our aim was not to focus on the number of diseases but to focus on any combination of two or more chronic diseases. We may say the contents were more relevant than the number.

Q9 - Lastly, in order to brush up on this study, the framework of meta-epidemiological methodology research may be helpful because all the studies included in this study were observational studies.

R9 – We appreciate the suggestion but considered the framework of meta-epidemiological methodology goes far beyond the scope of this review.

Reviewer 3 Report

This manuscript from Prof. Marjan van den Akke and colleagues review synthesizes studies to assess how multimorbidity affects the everyday lives of middle-aged persons, and identifies skills and resources that may help them overcome that burden.

The authors have integrated all assessment from 44 studies and found varying degrees of impact on global, physical, psychosocial and work functioning with multimorbidity among middle-aged persons. This review expects to help the healthcare professionals to provide holistic and dynamic health care plans based on actual needs of middle-aged persons with multimorbidity and probe their physical, financial, and interpersonal circumstances. Although this manuscript is well-designed and good writing, Some revisions are recommended as follows:

  1. “Education and Occupation” appears to be a critical characteristics especially under work functioning. Did the authors consider including it to the condition of patient’s characteristics?
  2. Studies are multi-regional, did the authors consider the different ADL with multimorbidity in terms of regionalization?
  3. Middle-age has two stage, Early Middle Age (ages 35–44) and Late Middle Age (ages 45–64). Did the authors have stratification analysis based on patient’s age?

Author Response

Revisions (R) made according to IJERPH reviewer's report (Reviewer 3) by queries (Q):

Q10 - This manuscript from Prof. Marjan van den Akker and colleagues review synthesizes studies to assess how multimorbidity affects the everyday lives of middle-aged persons, and identifies skills and resources that may help them overcome that burden. The authors have integrated all assessment from 44 studies and found varying degrees of impact on global, physical, psychosocial and work functioning with multimorbidity among middle-aged persons. This review expects to help the healthcare professionals to provide holistic and dynamic health care plans based on actual needs of middle-aged persons with multimorbidity and probe their physical, financial, and interpersonal circumstances. Although this manuscript is well-designed and good writing, some revisions are recommended as follows:

R10 - We would like to thank Reviewer 3 for the positive comments and the thorough reading of our manuscript.

Q11 - “Education and Occupation” appears to be critical characteristics especially under work functioning. Did the authors consider including it to the condition of patient’s characteristics?

R11 – Unfortunately, although we aimed to retrieve more variables to describe more accurately the characteristics of the patients included in the review, such as level of education, we were not able to extract such information due to underreporting of the included studies.

Q12 - Studies are multi-regional, did the authors consider the different ADL with multimorbidity in terms of regionalization?

R2 – Unfortunately, we did not consider the different ADL in terms of regionalization.

Q13 - Middle-age has two stage, Early Middle Age (ages 35–44) and Late Middle Age (ages 45–64). Did the authors have stratification analysis based on patient’s age?

R13 – We thank reviewer 3 for the suggestions but we did not stratify our sample by the two stages of middle age.

Round 2

Reviewer 2 Report

Thank you very much for your answer. I have read it carefully. I am worried that you have not answered my question properly and I am so sorry that I have to conclude that your paper should be rejected. Thank you very much for your understanding.

Author Response

Revisions (R) made according to IJERPH Reviewer 2 report by queries (Q):

We think that you adequately paid attention to most of the comments by the reviewers. However, reviewer 3 had some methodological questions. We would like to see some more reflection on these points in the discussion of the manuscript

Revisions (R) made according to IJERPH reviewer's report (Reviewer 3) by queries (Q):
Q1 - This manuscript from Prof. Marjan van den Akker and colleagues' review synthesizes studies to assess how multimorbidity affects the everyday lives of middle-aged persons and identifies skills and resources that may help them overcome that burden. The authors have integrated all assessments from 44 studies and found varying degrees of impact on global, physical, psychosocial, and work functioning with multimorbidity among middle-aged persons. This review expects to help the healthcare professionals to provide holistic and dynamic health care plans based on the actual needs of middle-aged persons with multimorbidity and probe their physical, financial, and interpersonal circumstances. Although this manuscript is well-designed and good writing, some revisions are recommended as follows:
R1 - We would like to thank Reviewer 3 for the positive comments and the thorough reading of our manuscript.
Q2 - “Education and Occupation” appears to be critical characteristics, especially under work functioning. Did the authors consider including it in the condition of the patient’s characteristics?
R2 – Unfortunately, although we aimed to retrieve more variables to describe more accurately the characteristics of the patients included in the review, such as level of education, we were not able to extract such information due to underreporting of the included studies.
Q12 - Studies are multi-regional, did the authors consider the different ADL with multimorbidity in terms of regionalization?
R2 – We did not observe different ADL in terms of regionalization. We added a sentence in the discussion section (lines 611-612) as follows:
“In general, we did not observe any pattern in terms of country or setting.”
Q3 - Middle-age has two-stage, Early Middle Age (ages 35–44) and Late Middle Age (ages 45–64). Did the authors have a stratification analysis based on the patient’s age?
R3 – We thank reviewer 3 for the suggestions but we did not stratify our sample by the two stages of middle age because the age groups in the papers were too dissimilar (and not accurate) to stratify according to early and late middle age. We have added the mean age of the population and the standard deviation into the key characteristics of the included studies (table 1) for a better description of the results. And we have also calculated the number of studies in which the mean age of the participants belonged to the early middle age and the late middle age and have included that information in table 2.